# Codon Usage Bias Analysis in the Chloroplast Genome of *Actinostemma tenerum* (Cucurbitaceae)

**DOI:** 10.3390/cimb47100833

**Published:** 2025-10-10

**Authors:** Jing-Jing Mu, Ji-Si Zhang

**Affiliations:** Liaoning Key Laboratory of Development and Utilization for Natural Products Active Molecules, Anshan Normal University, Anshan 114000, China; mujingjing@asnc.edu.cn

**Keywords:** *Actinostemma tenerum*, chloroplast genome, codon usage bias, optimal codon

## Abstract

The plant *Actinostemma tenerum* is endemic to East Asia and has been used as a traditional medicinal herb for over 1400 years. Investigating the chloroplast genome characteristics and codon usage bias (CUB) is essential for advancing research on molecular markers and genetic diversity in *A. tenerum*. In this study, we sequenced the complete chloroplast genome of *A. tenerum*, revealing a length of 160,579 bp, with a GC content of 36.5%. The genome comprised 132 coding genes, including 87 protein-coding genes (CDSs), 8 rRNA genes, and 37 tRNA genes. Analysis of the 51 selected CDSs showed average GC1, GC2, and GC3 values of 46.95%, 39.52%, and 28.11%, respectively. The effective number of codons (ENC) ranged from 35.34% to 56.23%, with an average of 45.57%, indicating a weak CUB. Nucleotide composition analysis revealed unequal distribution of A, T, C, and G, with codon preference biased towards A or U. Neutrality plots, ENC-plots, and PR2-bias plots indicated that natural selection predominantly influences on CUB. A total of 18 optimal codons were identified. This study contributes genetic insights into *A. tenerum* and enhances our understanding of codon usage patterns in plant chloroplast genomes.

## 1. Introduction

Chloroplasts are essential semi-autonomous organelles responsible for photosynthesis in plants, possessing their own genome [1]. The chloroplast genome is typically organized into a circular configuration, comprising a large single copy region (LSC), a small single copy region (SSC), and two inverted repeat sequences (IRs) [2,3,4]. The chloroplast genome holds significant application values for plant identification, assessment of genetic diversity, and study of gene expression [5,6,7,8]. With the decreasing cost of high-throughput sequencing, an increasing number of plant chloroplast genomes have been sequenced and annotated [9,10]. Comprehensive investigations of plant chloroplast genomes play a vital role in species taxonomy, phylogeny and conservation, while simultaneously shedding light on the evolutionary processes of plant lineages.

Codons, which link nucleotides to proteins, can encode a single amino acid through multiple synonymous codons, a phenomenon known as codon usage bias (CUB) [11]. CUB, prevalent in plants, reflects an adaptive selection mechanism for specific nucleotide combinations during transcription and translation [12]. This bias is influenced by multiple factors, including gene expression levels, GC content, natural selection, mutational pressure, and selective pressure, which collectively shape the patterns of codon preferences in plants [13,14,15]. Understanding CUB is crucial for revealing evolutionary relationships among species and optimizing gene expression efficiency.

*Actinostemma tenerum* Griff. (1845), a monotypic genus of the Cucurbitaceae, has long been utilized as a traditional medicine herb for approximately 1400 years. This species is endemic to East Asia and distributes across China, Indochina Peninsula, Korea and Japan. *A. tenerum* is an annual herb characterized by a tufted growth habit and creeping rhizome, with its entire plant being used for medicinal purposes and having been cultivated as an ornamental species in gardens [16]. Based on the morphological data, *Actinostemma lobatum* (Maxim.) Maxim. ex Franch. & Sav. was treated as synonymous with *A. tenerum* (https://www.plantplus.cn/ (accessed on 8 August 2024)). Phytochemical studies confirmed that this species contains saponins, polyphenols, lipids, and other compounds, especially 19 types of saponin were identified, which are similar to ginsenoside [17,18,19,20,21,22,23,24,25]. Additionally, the seeds of this species consisted of 11 fatty oil compounds by GC-MS analysis, with highest content of unsaturated fatty acid [24]. Further studies have proved that this species possesses several biological activities, such as anti-tumor, anti-bacterial, anti-inflammatory, and anti-thrombotic [26,27,28].

In this study, we utilized high-throughput sequencing technology to obtain the chloroplast genome of *A. tenerum* for the first time. Through assembly and annotation, we analyzed the chloroplast genome’s structure and examined its codon usage bias. Furthermore, we investigated the factors influencing the codon preference, aiming to provide a reference for organism evolution and help to understand the patterns of codons in chloroplast genomes in Cucurbitaceae.

## 2. Materials and Methods

### 2.1. Sampling, DNA Extraction and Sequencing

The samples were collected from Anshan City, Liaoning Province, China (41°6′26″ N, 122°59′20″ E). A voucher specimen (accession no. ZJS_2024075) was deposited in the specimen room of Anshan Normal University (https://www.asnc.edu.cn/ (accessed on 28 August 2024), Contact: Ji-Si Zhang, E-mail: zhangjisi@asnc.edu.cn). Genomic DNA was extracted from silica gel-dried leaves using the modified CTAB method [29], and an Illumina paired-end (PE) library was prepared and sequenced at Personalbio Biotechnology Co., Ltd., Shanghai, China.

### 2.2. Chloroplast Genome Assembly and Annotation

Raw sequencing reads were quality-filtered using Trimmomatic v.0.39 to remove adapter sequences and low-quality bases [30]. GetOrganelle v.1.5 was then employed to assemble the clean reads [31]. The chloroplast genome of *A. tenerum* was annotated using GeSeq (https://chlorobox.mpimp-golm.mpg.de/geseq.html (accessed on 2 January 2025)) and Geneious v.9.0.5 (http://www.geneious.com/ (accessed on 2 January 2025)) with *Gynostemma pubescens* (NC_036142) as the reference [32]. The annotated complete chloroplast genome of *A. tenerum* was deposited in GenBank under the accession number PV938953.

### 2.3. Calculation of Parameters Related to Codon Usage Bias

To minimize analytical errors, coding sequences shorter than 300 bp and repetitive coding sequences were excluded, and, thus, there were 51 CDSs selected for subsequent analysis [33]. CodonW v.1.4.2 and the Codon Usage Statistics Program (CUSP) tool from the European Molecular Biology Open Software Suite (EMBOSS) website (https://bioinformatics.nl/cgi-bin/emboss/cusp (accessed on 20 January 2025)) were employed to calculate the effective number of codons (ENC), relative synonymous codon usage (RSCU), and GC content at the 1st, 2nd, and 3rd positions of the codons (namely GC1, GC2, GC3, respectively) in the 51 CDSs of *A. tenerum*. Correlation analysis and significance testing were performed using SPSS v.29.0.

### 2.4. Neutrality Plot Analysis

A neutrality plot was constructed using GC3 and GC12 (the mean of GC1 and GC2). The plot was annotated with the regression equation and the coefficient of determination (*R*^2^) to assess the relationship between these variables. A regression coefficient nearing 1 signifies a predominant influence of mutational pressure on the gene, while a coefficient nearing 0 indicates that natural selection is the primary determinant affecting the gene [34,35].

### 2.5. ENC-Plot Analysis

An effective number of codons plot (ENC-plot) can reflect the extent to which codon preference is influenced by mutation and natural selection. A scatter plot correlating GC3 content with ENC values was constructed, alongside a fitted standard curve described by the equation ENC_exp_ = 2 + GC3s + 29/[GC3s^2^ + (1 − GC3s)^2^]. The standard curve serves as a reference boundary: ENC points above this curve suggest that mutational processes influence codon bias, indicating reduced impact of natural selection. Conversely, substantial deviations of points from the curve imply a stronger influence of natural selection pressure on codon bias [36].

### 2.6. PR2-Bias Plot Analysis

The ratios A3/(A3 + T3) and G3/(G3 + C3) were determined from the 51 CDSs of the *A. tenerum* chloroplast genome. A scatter plot was created with G3/(G3 + C3) and A3/(A3 + T3), with the center point marking A = T and C = G. Each point reflects the extent and direction of base deviation. Points in the upper half suggest a higher frequency of third-position codon A over T, while points in the left half indicate a higher frequency of C over G. If base mutations were the sole influence, base usage would be uniform, resulting in an even scatter distribution. Otherwise, it suggests codon preference is influenced by both mutation and natural selection [37,38].

### 2.7. Identification of Optimal Codons

Optimal codons are characterized by both high expression and frequency. The 51 CDS sequences of the *A. tenerum* chloroplast genome were ranked by ENC, and the top and bottom 10% were selected to form high-expression (*ycf3*, *ycf4*, *atpE*, *ycf2*, *rpl22*) and low-expression (*rps8*, *rps14*, *petD*, *rpl16*, *ndhJ*) gene libraries, respectively. CodonW 1.4.2 was used to calculate RSCU and ΔRSCU (ΔRSCU = RSCU_high_ − RSCU_low_) for these libraries. Codons with RSCU > 1 were deemed high-frequency, while those with ΔRSCU ≥ 0.08 were considered high-expression. Codons meeting both criteria were identified as optimal [33,36].

## 3. Results

### 3.1. Chloroplast Genome Characteristics of A. tenerum

A total of 6 Gb of 150 bp paired-end raw reads were utilized for the assembly of the chloroplast genome of *A. tenerum*. The chloroplast genome of *A. tenerum* spanned 160,579 bp, with a GC content of 36.5%. It displayed the typical quadripartite circular structure (Figure 1). The LSC region, encompassing 89,766 bp, harbored genes associated with photosynthesis and gene expression regulation. The SSC region with 18,553 bp predominantly contained genes involved in chloroplast function. Each IR region included 26,130 bp and maintained the stability of the chloroplast genome.

The chloroplast genome of *A. tenerum* comprised a total of 132 genes, consisting of 87 CDSs, 37 tRNA genes, and eight rRNA genes (Appendix A). Among these annotated genes, 15 genes contained a single intron, while three genes (*rps12*, *clpP*, *ycf3*) harbored two introns (Appendix A). These genes were classified into four main groups: 76 genes involved in self-replication, 46 genes related to photosynthesis, five genes with various functional roles, and five genes with uncharacterized functions. Notably, within the 46 photosynthesis-related genes, 18 genes encoded subunits of NADH dehydrogenase.

### 3.2. Codon Usage Patterns

Analysis of 51 CDSs selected from *A. tenerum* revealed a range of GC content from 31.23% (*ndhF*) to 46.04% (*rps11*) (Appendix A). Notably, significant differences in GC content were observed among the three codon positions (Figure 2), with average GC1, GC2, and GC3 values of 46.95%, 39.52%, and 28.11%, respectively (Appendix A). Nucleotide composition analysis demonstrated unequal distribution of A, T, C, and G, with a preference for codons ending in A or U. The ENC for most chloroplast genes exceeded 35, ranging from 35.34 to 56.22 (Appendix A). Specifically, six genes (*ndhJ*, *petD*, *psbA*, *rpl16*, *rps14* and *rps8*) exhibited ENC values below 40, while four genes (*atpE*, *ycf2*, *ycf3* and *ycf4*) had ENC values exceeding 50. The ENC values of the remaining genes fell within 40 to 50. These findings suggested a relatively weak codon bias in the chloroplast genome of *A. tenerum*.

Correlation analysis of codon parameters revealed that GC_all exhibited highly significant correlations with GC1, GC2, and GC3 (Table 1). GC1 and GC2 showed an extremely significant correlation, while neither correlated with GC3. These results suggested that the base compositions at the first and second codon positions were similar, but distinct from the third position. ENC values were statistically correlated with GC3, but not with GC1 and GC2. No correlations were observed between the number of codons and any parameters, indicating that gene sequence length does not influence GC content at different positions or the ENC value.

With the exception of the codons encoding Methionine and Tryptophan, 30 out of the remaining codons had RSCU values greater than 1 (Figure 3, Appendix A). Notably, all these codons terminated with either A or U, except for UUG. This finding further denoted that the chloroplast genome of *A. tenerum* has a preference for codons ending with A or T.

### 3.3. Neutrality Plot Analysis

In the chloroplast genome of *A. tenerum*, the GC12 values of 51 CDSs varied from 33.5% to 54.7%, while the GC3 values ranged from 21.0% to 36.8%. These findings indicated that there is a certain difference in the proportions of GC12 and GC3 in the chloroplast genome of *A. tenerum*. The regression analysis yielded a coefficient of 0.2027 for GC12 with respect to GC3, accompanied by an *R*^2^ value of 0.0219, suggesting an absence of a significant correlation between GC3 and GC12 (Figure 4A). These results implied that CUB in *A. tenerum* may be marginally influenced by mutational pressures, while the interplay of natural selection and other factors likely holds substantial importance.

### 3.4. ENC-Plot Analysis

The ENC-plot analysis revealed that the majority of genes exhibited a distribution below the standard curve (the theoretical distribution of the genes’ ENC values), with only a minority clustering near it (Figure 4B). Analysis of the ENC ratio frequency distribution highlighted discrepancies between observed and expected ENC values (Table 2). Notably, eight genes (15.7%) fell within the range of −0.05 to 0.05, indicative of higher mutation pressure than selection pressure influencing codon bias. The remaining genes (84.3%) fell into other intervals, indicating a predominant impact of selection pressure. These findings inferred that a significant influence of natural selection on codon usage bias in the chloroplast genome of *A. tenerum*.

### 3.5. Parity-Rule 2 (PR2) Bias Plot Analysis

The PR2-plot analysis highlights the differential usage of the bases A, T, G, and C at the third of codons. With selective pressure, mutations at these positions should occur randomly, leading to similar base frequencies. However, as shown in Figure 4C, the chloroplast genes of *A. tenerum* exhibited an uneven distribution across the four quadrants, with the majority concentrated in the lower half. Specifically, the G3/(G3 + C3) values of 31 genes exceeded 0.5, while the A3/(A3 + T3) values of 35 genes were less than 0.5, indicating a preference for T over A and G over C in the third position base usage (Appendix A). These findings suggested that natural selection primarily influenced the third base during the process of evolution.

### 3.6. Identification of Optimal Codons

The optimal codons for the CDSs of *A. tenerum* were presented in Table 3. Among the 30 high-frequency codons with an RSCU value exceeding 1, 29 of them terminated with either A or U (Appendix A). Additionally, 26 codons exhibited high expression, with a ΔRSCU value of at least 0.08. By analyzing the intersection of these two sets, there were 18 optimal codons identified in the chloroplast genome of *A. tenerum*, including GCA, AGA, CAA, GAA, GGA, GGU, CAU, AUA, AUU, UUA, UUG, AAA, UUU, UCU, ACA, UAU, GUA and GUU. Notably, 17 of these optimal codons ended with A or U. These results also suggested a preference for codons ending with A or U in CDSs of the chloroplast genome of *A. tenerum*.

## 4. Discussion

### 4.1. The Chloroplast Genome Characteristics of A. tenerum

In this study, the complete chloroplast genome of *A. tenerum* was firstly sequenced and annotated. The chloroplast genome had a size of 160,579 bp with a GC content of 36.5% (Figure 1), and displayed the typical quadripartite structure with other angiosperms [4,9]. It consisted of the largest LSC region (89,766 bp), SSC region (18,553 bp), and two IR regions (26,130 bp) (Appendix A). This result is consistent with the other Cucurbitaceae plants [39,40,41,42]. For instance, Zhang et al. [39] detected a comparative chloroplast genome characteristic in the Cucurbitaceae, and revealed that the ten genera exhibited a conserved quadripartite structure with similar region lengths, comprising an LSC region spanning 86,642 to 88,374 bp, an SSC region spanning 17,897 to 18,653 bp, and two IR regions ranging from 25,193 to 26,242 bp. Similarly, Jiang et al. [42] demonstrated that the chloroplast genome sizes of 11 *Trichosanthes* ranged from 156,413 to 157,556 bp with a uniform GC content of 37%. Specifically, within the *Trichosanthes*, the LSC region spanned 85,642 to 88,374 bp, the SSC region ranged from 17,897 to 18,653 bp, and the IR regions varied from 25,193 to 26,242 bp in length. Moreover, *A. tenerum* encompassed 132 annotated genes, including 87 CDSs, 37 tRNA genes and eight rRNA genes (Appendix A), which were highly similar to those of *Momordica charantia* and *Lagenaria siceraria* within the Cucurbitaceae, characterized by the absence of the *infA* gene and the retention of the *ycf1* gene [39]. In this study, the chloroplast genome of *A. tenerum* exhibited a slightly larger size compared to other genera in the Cucurbitaceae, primarily due to a slight expansion of the LSC region. Taken together, the complete chloroplast genomes of Cucurbitaceae are highly conserved in terms of size, structure, gene order and content. These characteristics are valuable for the exploration of genome divergences and the identification of selection signals throughout evolutionary history.

### 4.2. Codon Usage Patterns and Their Drivers

Codon usage bias serves as a crucial indicator for studying the evolutionary relationships among plant chloroplast genomes [12,43] and is closely associated with the GC content of codons within chloroplast genomes [13,44,45,46]. Analysis of codon usage patterns identified significant variations in GC content among the first, second and third codons within the 51 selected CDSs, with GC3 exhibiting the lowest GC content (Table 1). No significant correlation was observed between GC12 and GC3 (Table 1). Additionally, the ENC values of these CDSs ranged from 35.34 to 56.22, with the majority exceeding 35 (Figure 2), indicating relatively weak codon bias in the chloroplast genome of *A. tenerum*. Moreover, most codons in the chloroplast genome of *A. tenerum* terminated with either A or U (Figure 3, Appendix A). The similar weak codon usage bias and codon preferences were explored in other Cucurbitaceae species [42,47]. Previous studies revealed ENC values ranging from 55 to 56, with a preference for A and U at the third codon position on chloroplast genome genes in 11 *Trichosanthes* species [42]. Also, the genus *Gynostemma* demonstrated low codon usage bias and a similar preference for A and U at the third nucleotide position [47].

The formation of codon usage bias is a complex process influenced by natural selection, mutational pressures, genome size, and tRNA abundance [13,14,15]. A number of studies have indicated that natural selection and mutations are the primary drivers of CUB [13,44]. Analysis of *A. tenerum*’s CUB revealed a slight impact of mutational pressure, as indicated by the lack of significant correlation between GC3 and GC12 (Figure 4A). ENC-plot analysis of the 51 selected genes showed that the ENC ratio frequencies of 43 genes were distributed outside of the range −0.05 to 0.05 (Table 2), suggesting a significant influence of natural selection on the chloroplast genome’s CUB in *A. tenerum*. The PR2-plot illustrated uneven gene loci distribution in the chloroplast genome (Figure 4C), indicating the primary effect of natural selection on the third base during evolution. Overall, multivariate analysis identified natural selection as the primary influencing factor, followed by mutational pressure and other factors. These results align with studies in related angiosperms, such as the monocotyledons *Zingiber* [48] and the family Araceae [49,50], the basal angiosperm *Manglietia* [51], the important horticultural crop family Rutaceae [52] and *Cucumis sativus* within the Cucurbitaceae [53]. These results highlight the predominant role of natural selection in shaping codon usage bias, and are vital to explore the chloroplast genome evolution of angiosperms.

Additionally, eighteen optimal codons were identified based on their high frequency and expression levels. Notably, 17 of these optimal codons terminated with A or U. This pattern resembled the findings in other Cucurbitaceae genera. For instance, there were 12 optimal codons revealed by the *Gynostemma*, 11 of which ending with A or U [47]. Similarly, research on *Cucumis sativus* also showed a preference for A or U-ending optimal codons [53]. These consistent trends across different species underscore the universality and conservativeness of optimal codon usage.

## 5. Conclusions

Based on the assembly and annotation results, the chloroplast genome of *A. tenerum* was determined to be 160,579 bp in size, comprising the LSC, SSC, IRa, and IRb regions, exhibiting a quadripartite structure with an overall GC content of 36.5%. Additionally, the chloroplast genome of *A. tenerum* harbored 132 protein-coding genes. Notably, there were eighteen optimal codons, including GCA, AGA, CAA, GAA, GGA, GGU, CAU, AUA, AUU, UUA, UUG, AAA, UUU, UCU, ACA, UAU, GUA and GUU, all ending with A or U except for UUG. Analysis through Neutrality plot, ENC-plot and PR2-plot revealed that codon usage bias in the chloroplast genome of *A. tenerum* was mainly influenced by natural selection, with a preference for A and U bases. This study could offer insights into the evolution of *A. tenerum* and enhance our comprehension of codon patterns within angiosperm chloroplast genomes.

## Figures and Tables

**Figure 1 cimb-47-00833-f001:**
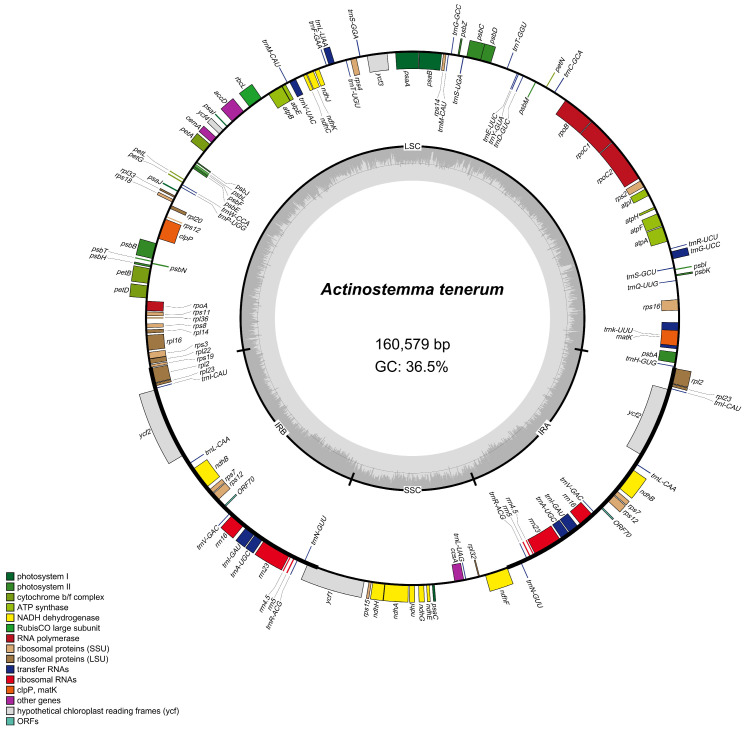
Circular map of the *A. tenerum* chloroplast genome. Genes transcribed counterclockwise were inside the circle, while those transcribed clockwise were outside. Colors differentiate gene functions. The inner circle’s dashed area showed GC content in dark grey and AT content in light grey. LSC: large single-copy region; IR: inverted repeat; SSC: small single-copy region.

**Figure 2 cimb-47-00833-f002:**
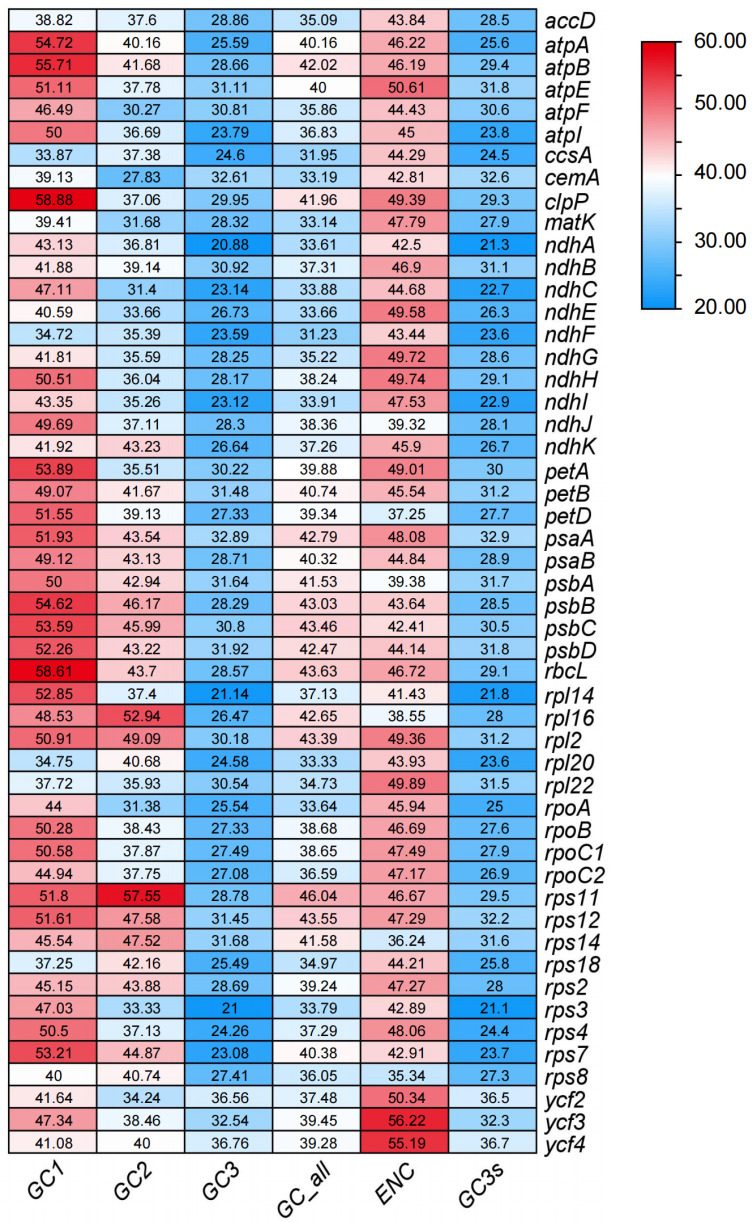
The heatmap of ENC in *A. tenerum*.

**Figure 3 cimb-47-00833-f003:**
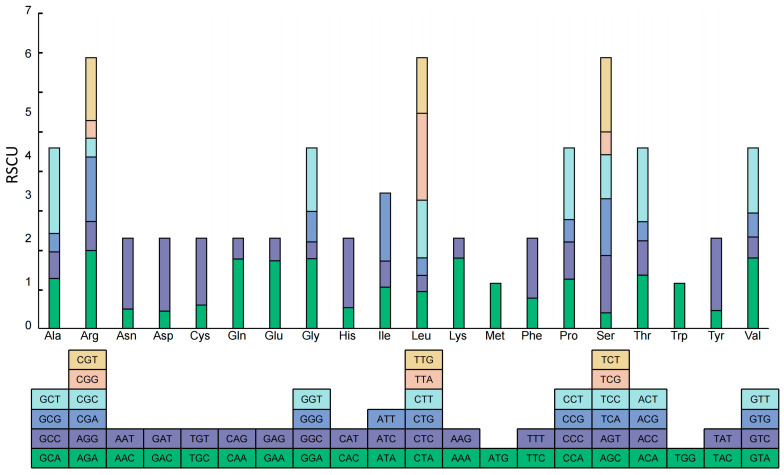
Codon content of 20 amino acids in all protein-coding genes of the *A. tenerum* chloroplast genome.

**Figure 4 cimb-47-00833-f004:**
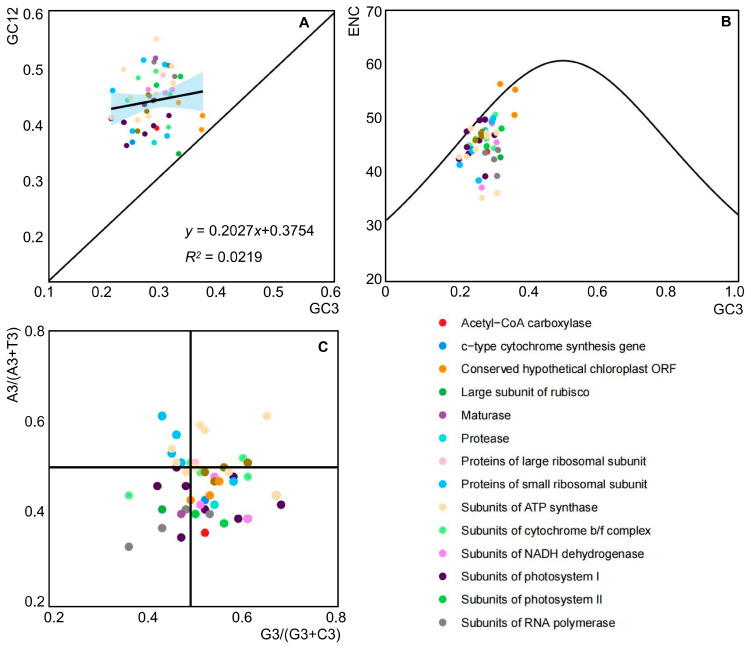
Plots of the causes of codon preference in the chloroplast genome of *A. tenerum*. (**A**) Neutrality plot analysis; (**B**) ENC plot analysis; (**C**) PR2-plot analysis.

**Table 1 cimb-47-00833-t001:** Correlation analysis of codon parameters in *A. tenerum* chloroplast genes.

Variation	GC1	GC2	GC3	GC_all	ENC
GC2	0.363 **				
GC3	0.090	0.159			
GC_all	0.787 **	0.775 **	0.461 **		
ENC	0.015	−0.190	0.349 *	0.025	
Codon No.	−0.028	−0.126	0.257	0.003	0.192

Note: ** *p* < 0.01, * *p* < 0.05.

**Table 2 cimb-47-00833-t002:** Distribution of ENC ratio frequency.

Class Range	Class Mid Value	Frequency Number	Frequency
−0.05~0.05	0	8	0.157
0.05~0.15	0.1	33	0.647
0.15~0.25	0.2	6	0.118
0.25~0.35	0.3	4	0.078
Total		51	1

**Table 3 cimb-47-00833-t003:** Optimal codons in chloroplast genome of *A. tenerum*.

Amino Acid	Codon	High-Expression Gene		Low-Expression Gene		∆RSCU
		RSCU	No.	RSCU	No.	
Ala	GCA **	1.54	15	1.05	25	0.49
Arg	AGA *	1.97	21	1.76	57	0.21
Gln	CAA *	1.53	13	1.38	80	0.15
Glu	GAA *	1.59	27	1.31	109	0.28
Gly	GGA *	1.78	24	1.65	56	0.13
	GGU **	1.41	19	1.06	36	0.35
His	CAU **	1.76	15	1.46	52	0.30
Ile	AUA *	1.19	25	1.02	84	0.17
	AUU *	1.33	28	1.24	102	0.09
Leu	UUA ***	2.06	22	1.09	55	0.97
	UUG *	1.59	17	1.47	74	0.12
Lys	AAA **	1.60	32	1.29	117	0.31
Phe	UUU **	1.48	17	1.03	99	0.45
Ser	UCU ***	2.05	14	1.52	70	0.53
Thr	ACA *	1.55	12	1.31	41	0.24
Tyr	UAU *	1.69	22	1.59	90	0.10
Val	GUA **	1.66	17	1.25	33	0.41
	GUU *	1.46	15	1.17	31	0.29

* 0.08 ≤ ∆RSCU < 0.30, ** 0.30 ≤ ∆RSCU < 0.50, *** 0.50 ≤ ∆RSCU.

## Data Availability

The genome sequence of this study was deposited in GenBank of NCBI (https://www.ncbi.nlm.nih.gov/ (accessed on 16 July 2025)) under accession no. PV938953.

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
