# Peer review of "Codon Usage Bias Analysis in the Chloroplast Genome of Actinostemma tenerum (Cucurbitaceae)"

_cimb, 2025, doi:10.3390/cimb47100833_

Round 1
Reviewer 1 Report
Comments and Suggestions for Authors
The article presents the data of bioinformatics analysis of the Actinostemma tenerum plastid genome sequence obtained by the authors for the first time. The obtained data themselves, both experimental and analysis results, are presented correctly and are of scientific interest.
However, simply presenting these data «as is» is not enough for a CIMB journal article. The manuscript barely discusses your results. Can you compare your findings for the Actinostemma tenerum plastid genome with those known for other species? Close and distant taxa? For plants growing in similar environmental conditions? Since such an analysis has not been performed, you cannot draw any conclusions about the adaptive significance of the characteristics found at this time. You wrote: “…reflected an adaptive strategy during its evolution, making it more suited to specific environmental conditions,” but you provide no suggestions about possible mechanisms (does this GC content correlates somehow with certain environmental conditions? Do you suggest that it might, and can you explain your suggestions?) or trends known from the literature.
I recommend expanding the Discussion chapter by analyzing the literature and comparing it with your results. Perhaps such an analysis will allow you to put forward some interesting hypotheses in the context of interpreting your results.
In addition, there are several technical comments on the manuscript:
- P. 9
The plant Actinostemma tenerum is endemic in East Asia and as a traditional medicine herb
Please correct the grammar.
- Ð . 10, 48
since Tang Dynasty
It would be better to indicate what time period the Tang Dynasty belongs to. The reader may not know this.
- 112-123
Using the ENC value as the selection criterion, the chloroplast genes of A. tenerum were ranked. The top and bottom 10% of genes were used to construct high and low pref- erence gene pools, respectively. The RSCU values were comprehensively measured, and the difference between the RSCU values of high and low expression genes, denoted as ΔRSCU, was calculated. Codons with ΔRSCU ≥ 0.08 and RSCU > 1 were identified as op- timal codons. The Materials and Methods should be described with sufficient details to allow others to replicate and build on the published results. Please note that the publica- tion of your manuscript implicates that you must make all materials, data, computer code, and protocols associated with the publication available to readers. Please disclose at the submission stage any restrictions on the availability of materials or information. New methods and protocols should be described in detail while well-established methods can be briefly described and appropriately cited.
A piece of instructions for authors ended up in the manuscript.
- 125, 217
Chloroplast genome characters of A. Tenerum
You mean characteristics?
- Ð . 233-235
Our results found that the GC content of codons in the chloroplast genome of A. tenerum is 36.5%, which reflected an adaptive strategy during its evolution, making it more suited to specific environmental conditions.
Additional research? Literature data that such a proportion of GC correlates with specific adaptation strategies?
- Please italicize all gene names, as well as all taxon and species names, throughout the manuscript.
Author Response
Reviewer 1
Q1: The article presents the data of bioinformatics analysis of the Actinostemma tenerum plastid genome sequence obtained by the authors for the first time. The obtained data themselves, both experimental and analysis results, are presented correctly and are of scientific interest.
However, simply presenting these data “as is” is not enough for a CIMB journal article. The manuscript barely discusses your results. Can you compare your findings for the Actinostemma tenerum plastid genome with those known for other species? Close and distant taxa? For plants growing in similar environmental conditions? Since such an analysis has not been performed, you cannot draw any conclusions about the adaptive significance of the characteristics found at this time. You wrote: “…reflected an adaptive strategy during its evolution, making it more suited to specific environmental conditions,” but you provide no suggestions about possible mechanisms (does this GC content correlates somehow with certain environmental conditions? Do you suggest that it might, and can you explain your suggestions?) or trends known from the literature.
Response 1: Thanks very much for your constructive comments and suggestions. We have revised as followed: (1) We have compared the findings of Actinostemma tenerum to other Cucurbitaceae species and other angiosperms in DISCUSSION (Page 9, Lines 224-242; Page 10, Lines 251-257, Lines 269-274). (2) As you pointed that it is improper that GC content correlated to certain environmental conditions without analysis, we have deleted the sentence “…reflected an adaptive strategy during its evolution, making it more suited to specific environmental conditions”. The findings in this study would provide a reference for organism evolution and help to understand the patterns of co-dons in chloroplast genomes in Cucurbitaceae.
Q2:I recommend expanding the Discussion chapter by analyzing the literature and comparing it with your results. Perhaps such an analysis will allow you to put forward some interesting hypotheses in the context of interpreting your results.
Response 2: Thanks for your professional suggestions. We have compared our results with other genera of Cucurbitaceae (Zhang et al. 2018, Molecules, 23, 2165; Shi et al., 2019, PLoS ONE, 14, e0226865; Bellot et al., 2020, Sci. Rep., 10, 488; Jiang et al., 2023, J. Anqing Norm. Univ., 29, 87–95.) The chloroplast genome of A. tenerum exhibited a slightly larger size compared to other genera in the Cucurbitaceae, because of a slight expansion of the LSC region. Generally speaking, the complete chloroplast genomes of Cucurbitaceae are highly conserved in terms of size, structure, gene order and content (Page 9, Lines 224-242). Especially, it was highly similar to that of Momordica charantia and Lagenaria siceraria in that the absence of the infA gene and the retention of the ycf1 gene (Page 9, Lines 234-236).
In addition, there are several technical comments on the manuscript:
Q3:P. 9 “The plant Actinostemma tenerum is endemic in East Asia and as a traditional medicine herb” Please correct the grammar.
Response 3: We have corrected it (Page 1, Line 9).
Q4: Ð . 10, 48
since Tang Dynasty
It would be better to indicate what time period the Tang Dynasty belongs to. The reader may not know this.
Response 4: Revised as your suggestion (Page 1, Line 10; Page 2, Line 46).
Q5: 112-123
Using the ENC value as the selection criterion, the chloroplast genes of A. tenerum were ranked. The top and bottom 10% of genes were used to construct high and low pref- erence gene pools, respectively. The RSCU values were comprehensively measured, and the difference between the RSCU values of high and low expression genes, denoted as ΔRSCU, was calculated. Codons with ΔRSCU ≥ 0.08 and RSCU > 1 were identified as op- timal codons. The Materials and Methods should be described with sufficient details to allow others to replicate and build on the published results. Please note that the publica- tion of your manuscript implicates that you must make all materials, data, computer code, and protocols associated with the publication available to readers. Please disclose at the submission stage any restrictions on the availability of materials or information. New methods and protocols should be described in detail while well-established methods can be briefly described and appropriately cited.
A piece of instructions for authors ended up in the manuscript.
Response 5: Thanks very much for your constructive comment. We have revised the M & M, and provided the detailed parameter and information for each analysis (see the updated M & M). Further, we have added two supplementary tables to support our results (Table S3 RSCU values reflecting codon usage bias in the chloroplast genes of A. tenerum. Table S4 The quadrant distribution data of each gene in the PR2-plot in the chloroplast genes of A. tenerum).
Q 6: 125, 217
Chloroplast genome characters of A. Tenerum
You mean characteristics?
Response 6: Revised as your suggestion (Page 9, Line 220).
Q7: Ð . 233-235
Our results found that the GC content of codons in the chloroplast genome of A. tenerum is 36.5%, which reflected an adaptive strategy during its evolution, making it more suited to specific environmental conditions.
Additional research? Literature data that such a proportion of GC correlates with specific adaptation strategies?
Response 7: Jiang et al. (2023, J. Anqing Norm. Univ., 29, 87–95) demonstrated that the chloroplast genomes sizes of 11 Trichosanthes was with a uniform GC content of 37%. Zhang et al. (2018, Molecules, 23, 2165) showed GC content ranging from 36.7% to 37.2% across ten genera in the Cucurbitaceae. The comparison implied that the GC content of the chloroplast genome of A. tenerum was a little lower than other genera, and it is interesting to explore the reason in future research.
Q8: Please italicize all gene names, as well as all taxon and species names, throughout the manuscript.
Response 8: Revised as your suggestion through the whole manuscript.

Reviewer 2 Report
Comments and Suggestions for Authors
The manuscript “Codon usage bias analysis in the chloroplast genome of Actinostemma tenerum (Cucurbitaceae)” by Mu and Zhang analyzed the complete chloroplast genome of Actinostemma tenerum, a traditional East Asian medicinal herb, to better understand its genetic characteristics and codon usage bias (CUB). The authors found the chloroplast genome to be 160,579 base pairs long with 135 genes. Analysis of 51 protein-coding genes revealed a weak CUB, with a preference for codons ending in A or U. The study's neutrality, ENC, and PR2-bias plots all suggest that natural selection is the primary factor influencing the CUB in this species, rather than mutational pressure. This research provides valuable genetic information for A. tenerum and contributes to the broader understanding of codon patterns in plant chloroplast genomes.
While the article presented holds promise, aspects of it would benefit from additional refinement to enhance their robustness and rigor. Specific areas for improvement are outlined below:
Introduction:
Lines 49-51: “A. tenerum is an annual herb with a tufted habit and a creeping rhizome [16], and has been developed as an orna-mental plant in gardens. Economically, the whole plant of A. tenerum are used medicinally [16]”. Restructure these two sentences into one and correct the grammatical error (are).
Lines 51-53: “Actinostemma lobatum (Maxim.) Maxim. ex Franch. & Sav. was treated as the synonymous of A. tenerum (https://www.plantplus.cn/)”. This information is not relevant and is not related to the subject of the study.
Line 39: “com-mon”. Please, correct it.
Line 44: “Under-standing”. Please, correct it.
Line 50: “orna-mental”. Please, correct it.
Materials and Methods:
I recommend that authors include the versions of all programs and databases used.
Lines 71-73: “Total genomic DNA was extracted from silica gel-dried leaves using the modified CTAB method [29] and illumina paired-end (PE) library was prepared and sequenced in the Nanjing Novogene Biotechnology Co., Ltd., China”. The authors should add a new section that describes the DNA extraction and Illumina sequencing procedures.
Lines 74-75: “In total, 6 Gb of 150-bp paired-end raw reads were generated and used for chloroplast genome assembly”. This information should go in the results section.
Lines 74-75: “Trimmomatic 0.39 was used to organize and trim overrepresented sequences for getting the clean reads”. Could you explain why the overrepresented sequences were eliminated? Is this correct, or does it refer to adapter sequences?
Results
Lines 126, 127, 146, 156, 168, 175, 177, 180, etc: “A. tenerum”. Please, put in italics.
Lines 167-169: “Except for UUG, all of these codons ended with A or U. This result further denoted that the chloroplast genome of A. tenerum has a preference for codons ending with A or T”. What is explained in this paragraph is not observed in Figure 3 but shown in Table 3. Could the explanation be improved to understand it better?
Comments on the Quality of English LanguageThe manuscript requires revision by a native language specialist to address grammatical errors and improve paragraph structure.
Author Response
Response to Reviewer 2
Comments and Suggestions for Authors
The manuscript “Codon usage bias analysis in the chloroplast genome of Actinostemma tenerum (Cucurbitaceae)” by Mu and Zhang analyzed the complete chloroplast genome of Actinostemma tenerum, a traditional East Asian medicinal herb, to better understand its genetic characteristics and codon usage bias (CUB). The authors found the chloroplast genome to be 160,579 base pairs long with 135 genes. Analysis of 51 protein-coding genes revealed a weak CUB, with a preference for codons ending in A or U. The study's neutrality, ENC, and PR2-bias plots all suggest that natural selection is the primary factor influencing the CUB in this species, rather than mutational pressure. This research provides valuable genetic information for A. tenerum and contributes to the broader understanding of codon patterns in plant chloroplast genomes.
While the article presented holds promise, aspects of it would benefit from additional refinement to enhance their robustness and rigor. Specific areas for improvement are outlined below:
Q1: Introduction:
Lines 49-51: “A. tenerum is an annual herb with a tufted habit and a creeping rhizome [16], and has been developed as an orna-mental plant in gardens. Economically, the whole plant of A. tenerum are used medicinally [16]”. Restructure these two sentences into one and correct the grammatical error (are).
Response 1: Thanks very much. Revised as you suggestion and corrected the grammars (Page 2, Lines 48-49).
Q2: Lines 51-53: “Actinostemma lobatum (Maxim.) Maxim. ex Franch. & Sav. was treated as the synonymous of A. tenerum (https://www.plantplus.cn/)”. This information is not relevant and is not related to the subject of the study.
Response 2: Thanks for your suggestion. The genus Actinostemma is a monotypic genus of the Cucurbitaceae, only including Actinostemma tenerum Griff. (1845). The previous phytochemical studies cited here have used the synonymous name Actinostemma lobatum (Maxim.) Maxim. ex Franch. & Sav. In order to avoid ambiguity, we preferred to retain this sentence.
Q3: Line 39: “com-mon”. Please, correct it.
Response 3: Revised as your suggestion.
Q4: Line 44: “Under-standing”. Please, correct it.
Response 4: Revised as your suggestion.
Q5: Line 50: “orna-mental”. Please, correct it.
Response 5: Revised as your suggestion.
Q6: Materials and Methods:
I recommend that authors include the versions of all programs and databases used.
Response 6: Revised as your suggestion.
Q7: Lines 71-73: “Total genomic DNA was extracted from silica gel-dried leaves using the modified CTAB method [29] and illumina paired-end (PE) library was prepared and sequenced in the Nanjing Novogene Biotechnology Co., Ltd., China”. The authors should add a new section that describes the DNA extraction and Illumina sequencing procedures.
Response 7: Revised as your suggestion (Page 2, Lines 72-79).
Q 8: Lines 74-75: “In total, 6 Gb of 150-bp paired-end raw reads were generated and used for chloroplast genome assembly”. This information should go in the results section.
Response 8: Revised as your suggestion (Page 3, Lines 125*126).
Q9: Lines 74-75: “Trimmomatic 0.39 was used to organize and trim overrepresented sequences for getting the clean reads”. Could you explain why the overrepresented sequences were eliminated? Is this correct, or does it refer to adapter sequences?
Response 9: We have re-organized this sentence (Page 2, Lines 72-73).
Q10: Results
Lines 126, 127, 146, 156, 168, 175, 177, 180, etc: “A. tenerum”. Please, put in italics.
Response 10: Revised as your suggestion.
Q11: Lines 167-169: “Except for UUG, all of these codons ended with A or U. This result further denoted that the chloroplast genome of A. tenerum has a preference for codons ending with A or T”. What is explained in this paragraph is not observed in Figure 3 but shown in Table 3. Could the explanation be improved to understand it better?
Response 11: Thanks very much for your comment. The data in Figure 3 and Supplementary Table S3 showed that the codons terminated with A or U (Page 5, Lines 162-165).
Q12: Comments on the Quality of English Language
The manuscript requires revision by a native language specialist to address grammatical errors and improve paragraph structure.
Response 12: Thanks for your suggestion. We have revised the grammars and paragraph structure by an English expert. We have marked the revised content and English usage in blue through the manuscript.
Round 2
Reviewer 1 Report
Comments and Suggestions for Authors
The authors have significantly revised the manuscript. A discussion of the results in light of the literature has been added, and the description of the methods and results has been revised. Unsubstantiated conclusions have been eliminated, and the results are presented more objectively. I believe the article is suitable for publication in its current form.